# Single variant, yet "double trouble": TSC and KBG syndrome because of a large de novo inversion

Victoria Rodrigues Alves Barbosa[1,2,3,*], Tatiana Maroilley[1,2,3,*], Catherine Diao[1,2,3], Leslie Colvin-James[2,3], Renee Perrier[2,3,†], Maja Tarailo-Graovac[1,2,3,†]

Despite the advances in high-throughput sequencing, many rare disease patients remain undiagnosed. In particular, the patients with well-defined clinical phenotypes and established clinical diagnosis, yet missing or partial genetic diagnosis, may hold a clue to more complex genetic mechanisms of a disease that could be missed by available clinical tests. Here, we report a patient with a clinical diagnosis of Tuberous sclerosis, combined with unusual secondary features, but negative clinical tests including *TSC1* and *TSC2*. Short-read whole-genome sequencing combined with advanced bioinformatics analyses were successful in uncovering a de novo pericentric 87-Mb inversion with breakpoints in *TSC2* and *ANKRD11*, which explains the TSC clinical diagnosis, and confirms a second underlying monogenic disorder, KBG syndrome. Our findings illustrate how complex variants, such as large inversions, may be missed by clinical tests and further highlight the importance of well-defined clinical diagnoses in uncovering complex molecular mechanisms of a disease, such as complex variants and "double trouble" effects.

## Introduction

Rare diseases (RDs) are mostly genetic in origin. It is not rare for patients to receive a clinical diagnosis of a well-defined syndrome—so far known to be caused only by variants in one or couple of genes—yet no or partial genetic diagnosis after multiple genetic tests failing to identify a pathogenic variant, or in case of recessive disease, finding only one variant. With improved analytical methods on sequencing datasets, it has been shown that genome sequencing may help to resolve genetic mechanisms in such cases (Wahlster et al, 2021; Maroilley et al, 2022).

Indeed, over the past decade, short-read and long-read sequencing technologies have become more affordable and more accurate, facilitating the exploration of the whole genome. In addition, new approaches such as optical genome mapping are now offering more options to resolve the genetic origin of unsolved RDs cases. Worldwide, large efforts are funding the implementation of such technologies for large rare disease patient cohorts (100,000 Genomes Project Pilot Investigators et al, 2021; Elliott et al, 2022). However, the diagnosis rate rarely reaches 50%, with still many patients going through an exhausting diagnostic odyssey. Genetic dataset reanalysis, in this case, can also be of utmost help and has been proven to greatly increase molecular diagnosis (Liu et al, 2019).

Patients with clinical diagnosis can present more/less severe phenotype or additional features, unrelated to the primary diagnosis. This phenotypic variability in rare conditions might be because of complex genetic mechanisms such as genetic modifiers (Rahit & Tarailo-Graovac, 2020), but also dual molecular diagnosis (Kurolap et al, 2016), for which next-generation sequencing with advanced downstream analyses is pivotal in resolving the genetic origin to deliver an accurate molecular diagnosis (Balci et al, 2017).

Dual molecular diagnosis refers to clinical diagnosis of more than one genetic disorder in an individual (Posey et al, 2017; Hannah-Shmouni et al, 2021). It is also known as "double trouble" in the literature (Donkervoort et al, 2013). They can often be mistaken for a new disorder but the advent of next generation sequencing has offered new ways to reach an accurate diagnosis (Tarailo-Graovac et al, 2016). Usually, the patient will present more severe or additional features to the primary disorder, leading to suspicion of a secondary condition. Dual diagnosis cases are either "blended," in which both diseases cause similar phenotypes, leading to overlapping features, or "composite," in which distinct symptoms from both conditions are present (Rosina et al, 2022).

They can originate from double homozygosity, especially in consanguineous families, as it was reported by Vona et al (2017) in a case of Ellis-van Creveld syndrome and hearing loss caused by two homozygous variants in *EVC2* and *COL11A2* (Vona et al, 2017). In addition, dual diagnosis can be caused by the disruption of two genes responsible for autosomal dominant conditions (Tarailo-Graovac et al, 2016; Posey et al, 2017; Rosina et al, 2022).

[1]Department of Biochemistry and Molecular Biology, Cumming School of Medicine, University of Calgary, Calgary, Canada   [2]Department of Medical Genetics, Cumming School of Medicine, University of Calgary, Calgary, Canada   [3]Alberta Children's Hospital Research Institute, University of Calgary, Calgary, Canada

Correspondence: maja.tarailograovac@ucalgary.ca; renee.perrier@albertahealthservices.ca
*Victoria Rodrigues Alves Barbosa and Tatiana Maroilley are equal first authors
†Renee Perrier and Maja Tarailo-Graovac are equal senior and co-corresponding authors

To date, reported dual diagnoses were described predominantly as originating from two distinct events. In most cases, only single nucleotide variants (SNVs) are involved. For instance, Tarailo-Graovac et al (2016) identified five cases of dual diagnosis in a cohort of individuals with inborn errors of metabolism by applying whole-exome sequencing, including in a patient carrying a homozygous missense variant in *GJB2* and compound heterozygous missense variants in *NPL*. Posey et al (2017) also applied whole-exome sequencing and uncovered 101 diagnostics related to two or more loci, with 90% of them because of multiple SNVs.

Combinations of a pathogenic indel and a pathogenic SNV have also been characterized as origin of dual diagnosis cases. Ong et al (2012) reported in a patient a 16-bp deletion in *DPYD* and a missense SNV in *GLB1* in a patient with dihydropyrimidine dehydrogenase deficiency and GM1 gangliosidosis. Copy number variants (CNVs; including aneuploidy, microdeletion or microduplication) have been also reported in dual diagnosis patients. In some cases, they were found to cause a disease, whereas another independent variant was causing the second condition. For instance, Wallis et al (2016) have reported a 0.5-Mb deletion in *NRXN1* associated with a missense SNV in *HOXA13* causing an atypical hand–foot–genital syndrome with developmental delay and Posey and colleagues have reported CNVs involved in 12% of their dual diagnosis cohort (Posey et al, 2017).

Chromosomal inversions were the first type of genetic variants to be studied, after being initially discovered in *Drosophila* (Sturtevant, 1917). Their impact could initially be considered as mild as there are no alterations in gene copy numbers, and in fact, human genome naturally carries multiple inversions. For instance, a 3.5-Mb inversion involving olfactory receptors on chromosome 8 was found present in 26% of healthy controls, having no clinical significance (Giglio et al, 2002). Pericentric inversions on chromosomes 1, 2, 3, 5, 9, 10, and 16 are amongst the most common in the human genome (Hsu et al, 1987). However, even though the DNA content may remain intact, the breakpoints of an inversion may cause serious gene disruptions, or impair transcription regulatory elements, leading to disease association (Feuk, 2010). Hemophilia A is one of the best-characterized examples, being caused by variants on chromosome X, which includes inversions present in 43% of patients (Lakich et al, 1993). In addition, heterozygous inversions can impair chromosomal recombination and lead to differential gene expression patterns. They can affect both meiotic and somatic recombination and are also capable of inducing phenotypic variability in human diseases (Nomura et al, 2018). The effect of inversions is intricate and still not fully understood for a variety of reasons (Feuk, 2010). First, they are relatively rare, and multiple patients carrying the same inversion are often scarce. This is usually overcome when inversion is affecting a gene that has a previous association with the disease. Second, rearrangements in DNA that do not cause any loss or gain of DNA sequence can hardly be detected with arrays (Feuk, 2010). However, the advancement of sequencing technologies, such as paired-end sequencing, allowed new strategies for the discovery of inversions (Tuzun et al, 2005).

Here, we describe an unprecedented case of a dual diagnosis of Tuberous sclerosis (TSC2; #613254; OMIM) and KBG syndrome (#148050; OMIM) because of one single genomic event (inversion). The identification of such unique molecular diagnosis was uncovered using short-read whole-genome sequencing (srWGS) and guided by clinical diagnosis of TSC.

# Results and Discussion

### Case description

In our CombGene research study—approved by the Conjoint Health Research Ethics Board at the University of Calgary (REB19-0245)—where we also include patients with well-defined clinical diagnoses, yet no genetic variants in known genes (Maroilley et al, 2022), we enrolled a female patient with a clear clinical diagnosis of TSC presenting with subependymal nodules and cortical tubers, refractory epilepsy and infantile spasms, bilateral astrocytic retinal hamartomas, multiple cardiac rhabdomyomas, bilateral renal angiomyolipas, hypomelanotic macules, angiofibromas, and moderate to severe intellectual disability (non-verbal). She had additional clinical features not typical for TSC; specifically, hypotonia, short stature (third centile), and dysmorphic features. TSC is an autosomal dominant disorder with highly variable clinical presentation. It is characterized by dermatologic lesions, cerebral cortex malformations, hamartomatous tumors in multiple tissues such as brain, heart, and kidneys, infantile spasms, autism spectrum disorder, and cognitive disability (Salussolia et al, 2019). TSC incidence in the population is of one case per 6,000–10,000 live births, affecting over two million people worldwide (Hallett et al, 2011). TSC is caused by loss-of-function variants in the TSC complex subunit 1 and 2 genes (*TSC1* and *TSC2*). Therefore, after a clinical diagnosis of TSC, gene-centric clinical genetic tests are usually performed. Usual clinical tests to detect SNVs and CNVs in coding region of *TSC1* and *TSC2* were conducted in our proband, but none of them has identified any variants of significance in either *TSC1* or *TSC2*. The tests included the following: (1) *TSC1* and *TSC2* panel sequencing; (2) dosage analysis; (3) microarray analysis; and (4) comprehensive epilepsy panel (379 nuclear + 37 mitochondrial genes).

### SrWGS and identification of the structural variants (SVs)

To further seek for the genetic origin of the condition, genome sequencing of the proband's blood-derived DNA was performed using the Illumina TruSeq PCR-free DNA library and the Illumina NovaSeq 6000. Singleton srWGS was analysed as in Maroilley et al (2022, 2023a) and as described in Methods. As no rare SNVs or indels were detected affecting the *TSC1* or *TSC2*, we expanded the gene-centric analyses to include more complex variants (Maroilley et al, 2021; 2023a; 2023b). We identified a single complex variant with a breakpoint in *TSC2* (Fig 1).

The variant is a heterozygous inversion with one of the breakpoints at the genomic location Chr16:2,115,464 (GRCh37), in the middle of *TSC2* in intron 15 (NM_000548), expected to completely disrupt the function of this gene (Fig 1). The second breakpoint is localized at Chr16:89,518,641 in the first intron of *ANKRD11* (NM_013275; Fig 1). The *ANKRD11* has been associated with a dominant condition known as KBG syndrome. "KBG" refers to the surname initials of the three families originally diagnosed with the

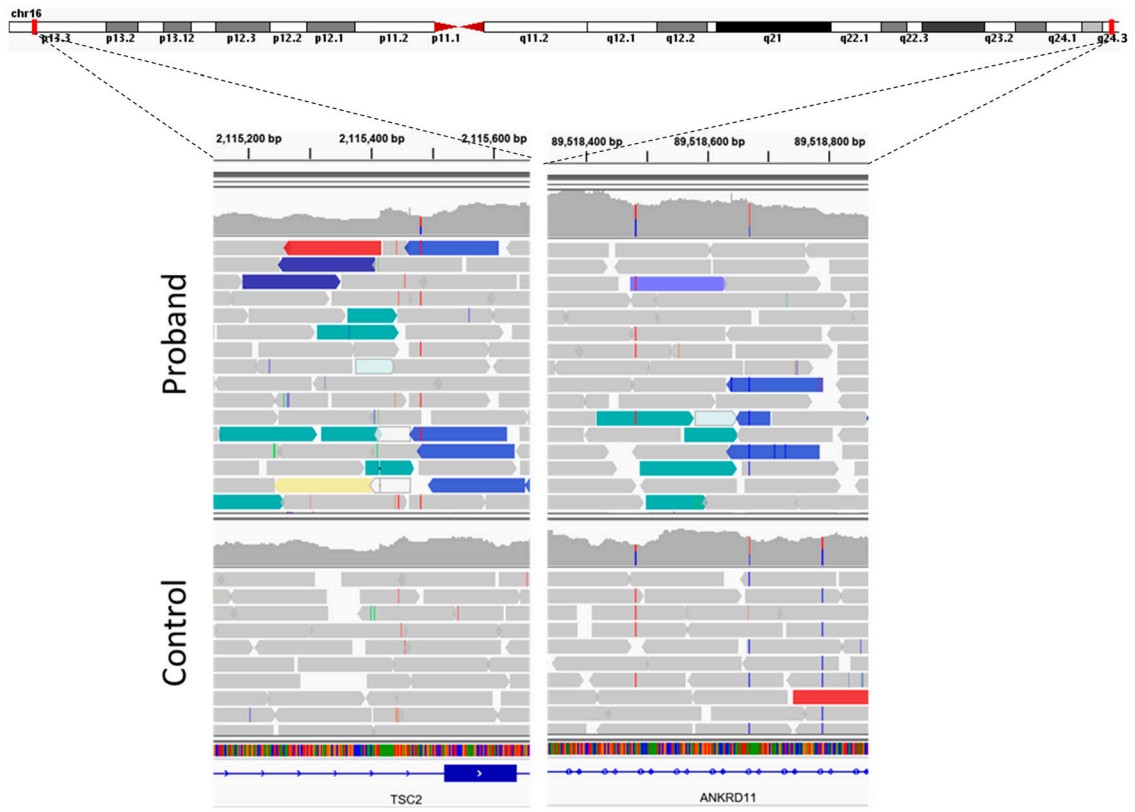

**Figure 1. Visualization with Integrative Genomics Viewer (IGV) of the reads aligned with the human reference genome (hg19) around the breakpoints' genomic locations in *TSC2* (left panel) and *ANKRD11* (right panel).**
On the middle track, for the proband, the reads are displaying a distinctive signature of a copy neutral inversion, compared with control. The reads colored in blue suggests that pair-mate reads were aligned on the same strand (+/+ or −/−), whereas they should be on opposite strand (+/−). The histogram on the upper track for each sample displays the coverage (number of reads aligned at each base). No large and continuous variation in coverage between the two breakpoints is observed, ruling out any rearrangement involving copy number variation. The lower track shows the annotation of the reference genome and help visualizing the intronic localization of both breakpoints.

syndrome (Herrmann et al, 1975). Therefore, our analyses revealed that the genetic cause of the condition in this proband is likely because of a complex genetic mechanism, an 87-Mb (87,403,177 bp; GRCh37) heterozygous pericentric inversion. To be certain that the inversion is the best possible explanation for the phenotype in this proband, we also performed unbiased singleton–WGS analysis where we considered entire genome and SNVs and SVs. Overall, after automatic filtration of polymorphisms (population frequency > 1%), low-quality variants and low effect variants, we manually filtered out 1,433 coding or splicing SNVs/indels (including 1,420 heterozygous and 13 homozygous) and 135 SVs based on population frequencies, predicted effect, pathogenicity, and gene–phenotype or gene–disease associations. This analysis resulted in the identification of single rare variant in a gene that matches phenotype of the patient; NC_000016.9:g. 2115464_89518641inv, the above noted inversion. Thus, our singleton–WGS analyses confirmed that there are no other variants of significance in the genome of this proband, but also that unbiased genome sequencing analysis can call and prioritize this large inversion from short-read sequencing data.

A brief review of ClinVar (https://www.ncbi.nlm.nih.gov/clinvar/) conducted in July 2022 revealed that over 60% of pathogenic and/or

likely pathogenic variants associated with human diseases are SNVs (105,407) and indels (3,298) (Fig S1A). The reported SVs are mostly CNVs, with 44,507 deletions and 3,298 duplications. In addition, 3,087 and 68 insertions and inversions, respectively, had been reported at the time. For *TSC1* and *TSC2* genes, about 40% of pathogenic or likely pathogenic variants are SNVs (186; 413) or indels (13; 22) (Fig S1A). In ClinVar, as in dbVar–repository dedicated to SVs larger than 50 bp (https://www.ncbi.nlm.nih.gov/dbvar/)—most of the pathogenic and/or likely pathogenic SVs are CNVs (Fig S1B). *TSC1* and *TSC2* are no exception, with only 156 and 60 insertions in ClinVar and 1 insertion in *TSC1* in dbVar. Our search in the literature revealed that there were only two balanced inversions reported for *TSC2* in TSC patients: a 600-kb inversion (Sampson et al, 1997) and a 550-kb inversion with breakpoints in introns 16 and 17 (Whitworth et al, 2018), neither nearly as large as the 87-Mb inversion identified in our patient.

### Analysis of the breakpoints and experimental confirmation of the inversion

The breakpoint in *TSC2* is 27 bp away from a SINE repetitive element (AluY; see Fig S2) and disrupts all isoforms of *TSC2* (Fig S2) either of

A

| srWGS sequence (read) | ACCTTTGAAGAGAAGGCCAGGACTTACACACAGGCTCATGATTCTGAGTCAGCAGCCGGGCAACAGAGTGAGACTCTGTCTCAAAACAAAACAA AAAAAAACTCATTTGCAATTTAGAGACAGGAATTACAGCCAAGTGCACCCCTTACCCC |
|---|---|
| Annotated srWGS sequence (read) | 16:2115521>//ACCTTTGAAGAGAAGGCCAGGACTTACACACAGGCTCATG(A/G)TTCTGAGTCAG**CAGCCG**//<16:6211544//16:89518635>//**CAGC C(G/T)**/<16:89518641//GGCAACAGAGTGAGACTCTGTCTCAAAA(C/T)AAAACAAAAAAAACTCATTTGCAATTTAGAGACAGGAATTACAGCCAAG TGCACCCCTTACCCC//<16:89518734 |

B

| | Breakpoint Junction 16p13.3 | Breakpoint Junction 16q24.3 |
|---|---|---|
| Sanger sequence | NNNNNNNNTGTCTCACGTCTATAATCCCAGAACTTTGGGAGGNCGA GGCAGGTGGATCACCTGACCTCAGGAGTTCAAGACCAGCCTGGAC AACATGGTGAAACCCCATCTCTACTAAAATACAAAAATTAACCAGGCA TGGTGGCACACGCCTATAATCCCAGCTACTCGGGAGGCTGCAGCA GGAGAATCGCTTGAACCTGGGAGGTGGAGGTTGCATTAAGCCGAC ATCACACCACTGCACTCCAGCCT**G**TGAAAACACTTTCTACCACAAAC ACCTTTTTTTTTTTTTTTTTTTTTTTGAAAACAAAATAAATACTAAANCG GGGGGGGGGGGGGNNGGGTGCCNGTAATCCCAGCACTTTGGGAGG CCCAACGNGGGTTAATTTTTGTCTGGGGAATCCCGACCATCCTGGN AACCACGGGGAAACCCCCTCCCCACCCCATAAANTATTTTTTTTTGG GTGTGGNGGGGGGCCCCTGTANTTTCNNCGGGGGNGAGTTAANC GGAGAATGGTGGGAACCTGGCAGGCGGCAATGAAAAANGGGGAG ATTGTGGCNNCCCCCCCNANCNGCGCGACANATAATCNNTN | NNNNNNGCNNGGGNNGACTCCAACACAACGCAGATGCAGCTGGTGGACG CTAGCCATGCGCGCCCCGGGAGCAGAGGCCTCGTGCAGAACACCCACCTG AAGGATGACCAGAAGCCCCAGGACGGCTGTCTTCACATCCTCCAAGGAGG CCGAGTATGCGGCCACATCCCTTTCTTCCAGCTCCGGGGGTGGGGAGAGG GAGCGGGCCATCACCTTTGAAGAGAAGGCCAGGACTTACACACAGGCTCAT GATTCTGAGTCAG**CAGCCG**GGCAACAGAGTGAGACTCTGTCTCAAAACAAA ACAAAAAAAAACTCATTTGCAATTTAGAGACAGGAATTACAGCNAAGTGCACC CNNNAGACCGTNGGCANNCCGNNAGGGNAGNAGGTGCGCTGGAGTGAAC CTGANGNGCGGGACGGNTTTCTCTCTACCNTGTCNGGNNTGACAGNNANG GAGNNTNTNNTNNTTGGAGGCGNNAGCGCCNCCCTCCNCCCTCCNCNTTG GGTGAGGNNNNGGGTGNCCCTTCACNANTNCGGACTTTTAAAGNTN |
| Annotated Sanger sequence | 16:89518397>//TGTCTCACGTCTATAATCCCAGAACTTTGGGAGGNC GAGGCAGGTGGATCACCTGACCTCAGGAGTTCAAGACCAGCCTGG ACAACATGGTGAAACCCCATCTCTACTAAAATACAAAAATTAACCAGG CATGGTGGCACACGCCTATAATCCCAGCTACTCGGGAGGCTGCAGC AGGAGAATCGCTTGAACCTGGGAGGTGGAGGTTGCATTAAGCCGA CATCACACCACTGCACTCCAGCCT**G**//<16:89518541//16:21154464>// **G**TGA(A/G)AACACTTTCTACCACAAACACCTTTTTTTTTTTTTTTTTTTTT TTTGA//<16:2115412 | 16:2115717>//GACTCCAACACAACGCAGATGCAGCTGGTGGACGCTAGCCAT GCGCGCCCCGGGAGCAGAGGCCTCGTGCAGAACACCCACCTGAAGGATG ACCAGAAGCCCCAGGACGGCTGTCTTCACATCCTCCAAGGAGGCCGAGTAT GCGGCCACATCCCTTTCTTCCAGCTCCGGGGGTGGGGAGAGGGAGCGGGC CCATCACCTTTGAAGAGAAGGCCAGGACTTACACACAGGCTCATG(A/G)TTC TGAGTCAG**CAGCCG**//<16:2115464////16:89518635>//**CAGCC(G/T)**//<16:89518 640//GGCAACAGAGTGAGACTCTGTCTCAAAACAAAACAAAAAAAAACTCATTT GCAATTTAGAGACAGGAATTACAGC//<16:89518715 |

**Figure 2.  Reads re-alignment for analysis of breakpoint junctions. (A)** Sequence of a split read for the srWGS data showing the breakpoint junction and realigned along the reference genome (hg19) using Blat UCSC. **(B)** Sequences of the breakpoints' junctions with Sanger sequencing, raw and annotated by alignment on hg19 using Blat UCSC.

intron 15, 12 or 14. The second breakpoint overlaps a SINE repetitive element (AluSx; see Fig S2) and disrupts all isoforms of *ANKRD11* (Fig S2). This suggests that this large pericentric inversion is SINE-mediated. One set of short split reads showed one of the breakpoints and the re-alignment of these reads along the reference genome (hg19) using Blat (Kent, 2002) showed no microhomology, but a hexamer CAGCCG that is present on both sides of the junction in the reference genome, with a substitution T>G (reference being CAGCCT) at the 16q24.3 locus (Fig 2A).

To confirm the inversion identified by srWGS, we realigned the reads around the breakpoints to identify split reads and guide the primer design for PCR confirmation (see the Materials and Methods section and Fig 3). We designed two primers to be located outside of the inversion (external forward primer P4 and external reverse primer P1) and two primers to be located inside of the inversion (internal forward primer P2 and internal reverse primer P3) (Fig 3A). Our PCR analyses revealed that whereas in all three family members and an unrelated control (Fig 3B) the WT sequence could be amplified (e.g., combination of P1 and P2 primers) (Fig 3B), the inverted sequence could only be detected in the affected proband (Fig 3B). Only the presence of the inversion, as predicted by the srWGS, would result in the proximity and correct orientation of the external and internal primers and therefore amplification of P3 (internal reverse primer) and P1 (external reverse primer) would be

possible (Fig 3A). Furthermore, Sanger sequencing of the band produced by the P1 and P3 primers in the proband confirmed the breakpoints as called by the srWGS data (Fig 3C). Importantly, our PCR-based assay showed that the inversion is de novo; only present in the affected proband but not in the unaffected parents, which agrees with the unremarkable family history. We analysed the breakpoint junctions using the Sanger sequencing (Fig 2B). The breakpoint junction sequence on 16p13.3 shows a blunt fusion with an overlap of one nucleotide: a G that is present on both sides of the breakpoint in the reference genome. The breakpoint junction sequence on 16q24.3 showed a blunt fusion and no microhomology and confirmed the presence of the hexamer already observed in the srWGS read sequence (Fig 2).

### Single event and dual diagnosis

To date, only two genes have been described to cause TSC: *TSC1* and *TSC2*. The spectrum of causal genetic variations so far reported in ClinVar and dbVar encompasses SNVs, indels small and large CNVs and small insertions and inversions (Fig S1). However, similar to our proband, 5–25% of the TSC patients remain without a genetic diagnosis after clinical genetic tests (Sancak et al, 2005; Northrup et al, 2013; Reyna-Fabián et al, 2020; Rosengren et al, 2020; Meng et al, 2021).

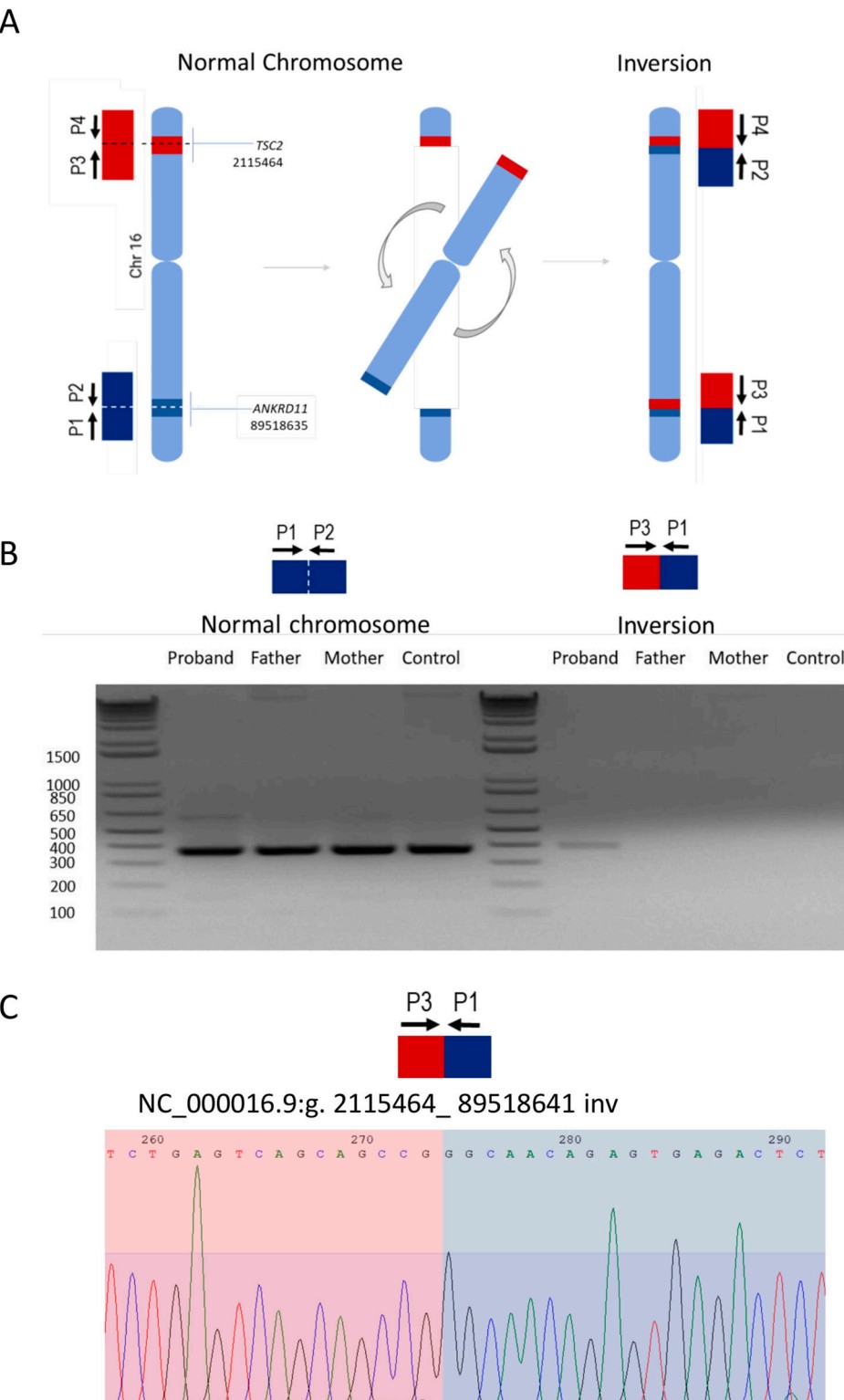

Figure 3. Breakpoints schematization and validation strategy for detection of the inversion. (A) Schematic representation on how this large pericentric inversion on chromosome 16 affects *TSC2* and *ANKRD11*. (B) PCR strategy for validation of the variant and confirmation of unaffected father and mother and affected heterozygote proband (de novo mutation) and PCR gel of gDNA amplification of an unrelated healthy control, father, mother, and proband. Left: detection of WT *ANKRD11*; Right: detection of Chr16:89,518,635 breakpoint in the proband using primers P1+P3. Expected variant band size for proband: 391 bp. (C) Sequence obtained at the junction breakpoint by Sanger using P3 primer for proband sample (5′-CGCACCCTCAGCAAATCCAG-3′).

Indeed, like other RDs, genetic diagnosis of TSC suffers from missing heritability (Maroilley & Tarailo-Graovac, 2019), when the genetic origin, although certain, cannot be found because of limited access to genetic testing or challenges in detection of complex genetic mechanisms such as mosaicism (Qin et al, 2010), intronic variants affecting splicing (Nellist et al, 2015), changes in promoter regions (van den Ouweland et al, 2011) or even complex variants like large genomic rearrangements (Eussen et al, 2000; Boehm et al,

2007; Kozlowski et al, 2007; Thibodeau et al, 2020). As SVs are known to contribute more to the human genomic variation (Pang et al, 2010), one could expect that they could be the cause of some of the genetically unsolved TSC patients. But those more complex genomic scenarios are overlooked and understudied as their detection sometimes necessitates a combination of expensive tests and technologies with advanced and time-consuming bio-informatics analyses. Our work shows that undetected balanced events such as inversions could contribute to the missing heritability in TSC cases.

Indeed, genome sequencing has the potential to facilitate the detection of complex genetic scenarios guided by well-defined clinical phenotypes (Nellist et al, 2015; Tyburczy et al, 2015; Redin et al, 2017). The ability of long reads to overcome the limitations of short reads in the detection of large events in the genome and to detect missing variants in patients with clinical diagnoses has been reported (Miller et al, 2021). Still, we and others have shown in model organisms (Itani et al, 2015; Tarailo-Graovac et al, 2015; Maroilley et al, 2021, 2023a) and in humans (Tarailo-Graovac et al, 2017; van Kuilenburg et al, 2018, 2019; Duvdevani et al, 2020; Grochowski et al, 2021; Zenagui et al, 2022) that short reads with advanced bioinformatics analyses have great potential in detecting and resolving a spectrum of genomic rearrangements. The proband we report here further supports our previous findings that srWGS could detect variants as complex as this 87-Mb inversion when advanced bioinformatics approaches are applied. In this case, even though inversions are often due to repetitive regions and the known limitations of srWGS in properly genotyping those regions, a clear signature of inversion could be detected based on split reads at the breakpoint junctions and pair-mates aligned in irregular orientations. Such signatures can either be detected by manually exploring read alignments on a genomic viewer, but several tools are now able to report it. By improving genomic pipelines with tailored bio-informatics approaches, srWGS represents a great opportunity to detect such variants.

Pericentric inversions, in particular, have been described in cases of infertility (Balasar et al, 2017; Sismani et al, 2020) and acute myeloid leukemia (Zhang et al, 2021), but also genetic syndromes such as Cri-du-chat syndrome (Faed et al, 1972), Duchenne muscular dystrophy (Shashi et al, 1996; Zaum et al, 2022) or Hermansky–Pudlak syndrome 2 (Jones et al, 2013). However, we did not find in the literature previous reports of pericentric inversion causing two autosomal dominant conditions. Mizuguchi et al (2021) reported monozygotic twin probands with syndromic intellectual disability caused by a paracentric 12-kb inversion disrupting two consecutive genes: *CPNE9* and *BRPF1*. Vaché et al (2020) described a paracentric 4.6-Mb inversion causing two fusion transcripts *PCDH15-LINC00844* and *BICC1-PCDH15* in a patient with Usher syndrome type 1. For TSC in particular, Sampson et al (1997) found a ~600-kb inversion with the proximal breakpoint within *TSC2* and others have reported inversions as part of more complex genomic rearrangements (Dabora et al, 2000; Eussen et al, 2000; Kozlowski et al, 2007; Thibodeau et al, 2020).

Our analysis of the whole genome and the characterization of the inversion at base-pair resolution uncovered that in addition to *TSC2*, this inversion likely disrupts the *ANKRD11* gene, responsible

for the autosomal dominant KBG syndrome. This unexpected finding could explain the presence of additional clinical features in our patient which were not felt to be in keeping with her diagnosis of TSC. These additional findings (including short stature, dysmorphic features, large central incisors, brachydactyly, and history of hypotonia) are in keeping with KBG syndrome supporting the likely functional impact of the inversion on the *ANKRD11* gene, and dual diagnosis of TSC/KBG syndrome.

To the best of our knowledge, this is the first report of a dual TSC/KBG syndrome because of a single SV, even though dual diagnosis cases for TSC or KBG have already been reported. Kopadze et al (2021) have reported a clinical dual diagnosis case of TSC and basement membrane disease. They found two compound heterozygous variants in *COL4A4* (p.Gly774Arg and p.Gly1465Asp) but no variant in either *TSC1* or *TSC2*. Localized next to *TSC2* on the genome, the *PKD1* gene is known to cause polycystic kidney disease when mutated. Thus, Kacerovska et al (2009) and Sampson et al (1997) have characterized patients with TSC/autosomal dominant polycystic kidney disease, because of deletions overlapping both *TSC2* and *PDK1* genes. Eussen et al (2000) have reported a familial translocation t(8; 16)(q24.3; p13.3) causing multiple diagnoses of TSC, adult polycystic kidney disease and hypomelanosis of Ito with $\alpha$-thalassemia trait.

Cianci et al (2020) have described a case of KBG syndrome and neurofibromatosis type 1 because of heterozygous frameshift mutation in *NF1* (p.Lys2401fs) and another in *ANKRD11* (p.Phe904fs). Tanaka et al (2021) described a woman with KBG syndrome and Tubulointerstitial kidney disease caused by a indel in *ANKDR11* (c.1903_1907del) and a SNV in *UMOD* (c.796T > C).

In sum, we here report the first case of dual diagnosis of TSC/KBG syndrome because of one single event, a large pericentric inversion on chromosome 16 that disrupts *TSC2* and *ANKRD11*, uncovered with srWGS. Our study demonstrates that for cases with clinical diagnosis of well-established genetic conditions, complete and accurate genetic diagnostic should be sought using genome sequencing as complex genetic mechanisms could be uncovered. Such discovery would greatly help to assess methods to better detect such genetic variants in patients with complex clinical presentation and reduce the diagnosis odyssey.

# Materials and Methods

### SrWGS

To search for the missing variant in either *TSC1* or *TSC2*, the family was enrolled in the COMBGENE research study, which was approved by the Conjoint Health Research Ethics Board at the University of Calgary (REB19-0245). Written informed consent was obtained for all participants. Using blood-derived DNA, genome sequencing of the proband was performed at The Hospital for Sick Children (SICKKIDS) using the Illumina TruSeq PCR-free DNA library and the Illumina NovaSeq 6000 with a targeted coverage of 40x. The median depth of coverage achieved was 50x.

## Bioinformatics analysis

The first stage of data analyses for patients with well-defined clinical features is gene-centric (Maroilley et al, 2022), guided by a hypothesis and focused on a single locus (Shu et al, 2023). After quality check with FAST-QC (Andrews, 2010) and trimming with Trimmomatic v0.35 (Bolger et al, 2014), reads were aligned along the human reference genome GRCh37 with BWA-MEM v0.7.15 (Li, 2013 *Preprint*), sorted by coordinates using Samtools v1.3.1 (Li et al, 2009) and marked for duplicates using MarkDuplicates Picard tool (Picard Tools - By Broad Institute, 2023).

We first called SNVs and indels using Freebayes v1.3.1 (Garrison & Marth, 2012 *Preprint*) and we looked for rare potentially pathogenic variants in both coding and noncoding regions of the loci of interest (*TSC1* and *TSC2*). We then scrutinized the loci by performing a visual assessment using Integrative Genomics Viewer (Robinson et al, 2011). We looked for signatures of short reads, combining split reads and/or changes in coverage that would suggest the presence of breakpoints of SVs or complex rearrangements disrupting either the *TSC1* or *TSC2*. We realigned the reads around the breakpoint identified in *TSC2* to uncover the structure of the variants and identify split reads that would guide the design of primers for PCR confirmation.

In addition, we performed unbiased genome-wide singleton-WGS analyses to exclude any potential variants not in *TSC1* or *TSC2* that could explain the phenotype; both de novo and inherited variants were considered, and coding/noncoding SNVs and SVs using in-house designed pipelines (Maroilley et al, 2022, 2023a). In brief, we semi-automatically filtered SNVs and indels called by Freebayes according to population frequencies, predicted effect on genes, and multiple pathogenicity scores as in Maroilley et al (2022). We also called SVs, complex genomic rearrangements, and mobile element insertions using RUFUS (Ostrander et al, 2018) and GRIDSS v.2.10.2 (Cameron et al, 2017). We used in-house developed scripts to filter and combine calls from both tools as in Maroilley et al (2023a). We called repeat expansions with ExpansionHunter (Dolzhenko et al, 2019) and ExpansionHunter Denovo v.0.9.0 (Dolzhenko et al, 2020) as in Maroilley et al (2023b). Finally, we explored large copy number variations using an in-house developed method based on the analysis of the coverage. The breakpoints of the inversion were further explored using the UCSC Genome Browser (Kent et al, 2002).

## Sanger sequencing

Testing was conducted using unrelated healthy control, unaffected father, unaffected mother, and affected proband gDNA samples. Two pairs of primers were designed targeting both breakpoints: primer 3 (P3) and primer 4 (P4) aligned to *TSC2*, whereas primer 2 (P2) and primer 1 (P1) targeted *ANKRD11* (Fig 2). The primer sequences are as follow: P1 5'-GGTGCACTTGGCTGTAATTC-3'; P2 5'-AACCTCTGGTATCAAATAGCTCA-3'; P3 5'-CGCACCCTCAGCAAATCCAG-3'; P4 5'- TGCACAGTCACTCGGGTATAAAGG-3'. As depicted in Fig 2, we used P1+P3 with annealing temperatures of 63°C. For amplification, High-Fidelity Phusion polymerase (NEB) was used, and PCR products were checked on a 2% agarose gel. The gel bands of interest were dissected from the gel and purified using the GFX PCR DNA and Gel Band Purification Kit (Cytiva). Purified DNA samples were then submitted for Sanger sequencing at the University of Calgary DNA Sequencing Facility. The sequences were then aligned on the human reference genome GRCh37 with Blat (https://genome.ucsc.edu/cgi-bin/hgBlat) to confirm the presence of the breakpoints. A part of the sequences obtained by Sanger aligned around the *TSC2* breakpoint, whereas the second part aligned on the opposite strand at the *ANKRD11* breakpoint.

## Databases and literature reviews

To report the distribution of pathogenic and/or likely pathogenic variants in both ClinVar (https://www.ncbi.nlm.nih.gov/clinvar/) and dbVar (https://www.ncbi.nlm.nih.gov/dbvar/) databases, the datasets were downloaded through their respective FTP sites (July 2022 releases). We included "Pathogenic," "Likely Pathogenic," and "Pathogenic/Likely Pathogenic" variants. We ignored "Micro-satellites" variants from ClinVar. To report the distribution of pathogenic and/or likely pathogenic variants reported for *TSC1* and *TSC2*, we first used the online version of each database, as the VCFs do not report all genes overlapped by a large genomic rearrangement. We obtained the ID of each variant from the dbVar website after applying appropriate filters for pathogenicity and gene, to then extract the precise characterization of each CNVs (deletion or duplication) from the VCF file.

# Data Availability

The datasets for this article are not publicly available because of concerns regarding participant/patient anonymity. Requests to access the datasets should be directed to the corresponding authors. The variant has been submitted to the "Global Variome shared LOVD" (DB_ID: TSC2_004723).

# Supplementary Information

# Acknowledgements

The authors thank the family for their participation in the research study and acknowledge APL (Alberta Precision Labs) and MDL (Medical Diagnostic Labs) for sample collection. This research was enabled in part by support provided by the Research Computing Services group at the University of Calgary. The authors received funding from the Alberta Children's Hospital Foundation, the Clinical Research Fund, CIHR-Bridge grant number OGB-185746, and an Alberta Children's Hospital Research Institute Postdoctoral Fellowship.

## Author Contributions

V Rodrigues Alves Barbosa: formal analysis and writing—original draft.

T Maroilley: formal analysis, funding acquisition, and writing—original draft.

C Diao: formal analysis and validation.

L Colvin-James: investigation, project administration, and writing—review and editing.

R Perrier: conceptualization, investigation, and writing—original draft.

M Tarailo-Graovac: conceptualization, formal analysis, supervision, funding acquisition, and writing—original draft.

## Conflict of Interest Statement

The authors declare that they have no conflict of interest.

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
