## [Reviewer comments · Life Science Alliance]

Life Science Alliance

Single variant, yet "double trouble": TSC and KBG syndrome due to a large de novo inversion

Victoria Rodrigues Alves Barbosa, Tatiana Maroilley, Catherine Diao, Leslie Colvin-James, Renee Perrier, and Maja Tarailo-Graovac

DOI: <https://doi.org/10.26508/lsa.202302115>

Corresponding author(s): Maja Tarailo-Graovac, University of Calgary and Renee Perrier, University of Calgary

Review Timeline:

Submission Date:	2023-04-26
Editorial Decision:	2023-06-19
Revision Received:	2023-12-15
Editorial Decision:	2023-12-18
Revision Received:	2024-01-12
Accepted:	2024-01-16

Transaction Report:

June 19, 2023

Re: Life Science Alliance manuscript #LSA-2023-02115-T

Dr. Maja Tarailo-Graovac
University of Calgary
Biochemistry & Molecular Biology
Calgary
Canada

Dear Dr. Tarailo-Graovac,

Thank you for submitting your manuscript entitled "Single variant, yet "double trouble": TSC and KBG syndrome due to a large de novo inversion" to Life Science Alliance. The manuscript was assessed by expert reviewers, whose comments are appended to this letter. We invite you to submit a revised manuscript addressing the Reviewer comments.

Thank you for this interesting contribution to Life Science Alliance. We are looking forward to receiving your revised manuscript.

Sincerely,

B. MANUSCRIPT ORGANIZATION AND FORMATTING:

Reviewer #1 (Comments to the Authors (Required)):

Authors Rodrigues Alves Barbosa Victoria et. al. reports a patient with a clinical diagnosis of Tuberous Sclerosis (TSC) and no molecular diagnosis from standard of care genetic testing. They performed short-read whole genome sequencing to identify an 87Mb pericentric inversion with breakpoints disrupting two known autosomal dominant disease genes TSC2 and ANKRD11.

This work is important in highlighting the challenges and limitations in the current standard of care genomic testing and the potential for finding novel variants expanding the mutational spectrum of previously known disease genes.

Comments to authors:

Authors mention that they first performed a targeted analysis in the known genes for TSC and later an unbiased analyses of all SNVs and SVs (page 7) that resulted in a single variant relevant to patient's phenotype which is the same inversion. It will be useful to describe and discuss the prioritization workflow and the number of variants they analyzed to get to a single variant explaining the phenotype. Large inversions are challenging to detect given the rate of false positives from short-read sequencing data. This seems to be a SINE-mediated inversion (both breakpoints seem to fall in or near an Alu repeat) and the structural variants from short-read sequencing is known to be enriched for false positives involving repeat elements. Discussion around this will help the readers appreciate the challenges and opportunities in the improving the genome sequencing based diagnostic tests.

Reviewer #2 (Comments to the Authors (Required)):

Summary:

Herein Victoria and Maroilley et al., present a case of a patient with an atypical presentation of Tuberous sclerosis (TSC). Short-read WGS uncovered a large 87 Mb pericentric inversion with breakpoints mapping to the genes TSC2 (explaining the TSC diagnosis) as well as ANKRD11 causing KBG syndrome suggesting a dual molecular diagnosis for this patient. Genomic inversions are a particularly cryptic form of structural variation as there is (typically) no copy number change associated with the event. Furthermore breakpoints may lie within Repetitive or other difficult to away regions. This work is very well written and presents an important point that atypical presentations of disease may suggest a second molecular "hit" and more through sequencing methodologies (such as short-read WGS) may be required to uncover the full picture.

The conclusions regarding the molecular diagnosis of TSC and KBG is supported by breakpoint junction alignment showing disruption of both genes.

Major Comments

I would add more information relating to genomic inversion into the introduction to give the reader a better view of how this type of rearrangement is associated with human disease states. See: Feuk, Lars. 2010. "Inversion Variants in the Human Genome: Role in Disease and Genome Architecture." *Genome Medicine* 2 (2): 11.

Posey et al., should be cited first as this was the major dual diagnosis paper as this (as far as I am aware) presents the concept more broadly to the field.

"However, the diagnosis rate rarely reaches 50%, with still many patients going through an exhausting diagnostic odyssey" - Consider citing the following: Liu, Pengfei, Linyan Meng, Elizabeth A. Normand, Fan Xia, Xiaofei Song, Andrew Ghazi, Jill Rosenfeld, et al. 2019. "Reanalysis of Clinical Exome Sequencing Data." *The New England Journal of Medicine* 380 (25): 2478-80. Which shows that reanalysis can be a useful tool even for unresolved patients.

"Dual molecular diagnosis, also known as "double trouble" in the literature, refers to clinical diagnosis of more than one genetic disorder in an individual." I have not come across this term "double trouble" in the literature and would stick to "dual molecular diagnosis".

Complex rearrangements involving multiple inversions have also been associated with other Mendelian disease: Grochowski, Christopher M., Ana C. V. Krepischi, Jesper Eisfeldt, Haowei Du, Debora R. Bertola, Danyllo Oliveira, Silvia S. Costa, James R. Lupski, Anna Lindstrand, and Claudia M. B. Carvalho. 2021. "Chromoanagenesis Event Underlies a de Novo Pericentric and Multiple Paracentric Inversions in a Single Chromosome Causing Coffin-Siris Syndrome." *Front. Genet.* 12: 1478.

What structural variant callers were run? Are there other CNVs that may be present? Was a chromosomal microarray or karyotype ever performed for this patient?

Figures:

Figure 1: it would be useful to include a depiction of the full chromosome 16 above the IGV view to show the exact regions that you are zooming into.

Figure 2: feels unnecessary for a main text figure and would be better suited for a supplemental figure.

Figure 3: this is a very informative and well laid out figure. It may be helpful to show the nucleotide level resolution of the breakpoints. Was it a blunt fusion? Microhomology? Are there genomic features in that region that may play a factor (Alus, LCRs, etc) in the formation of this inversion?

Was any experimental validation performed to show disruption of these genes? A qPCR of cDNA for these genes should show an expression difference from normal controls if they are being disrupted by the inversion. A western blot would also nicely show a disruption of the genes.

Minor Comments:

I don't see "KBG" syndrome spelled out in the manuscript.

Please include Chr when mentioning genomic locations so "Chr16:2,115,464"

"The second breakpoint is on the opposite strand..." I don't think "the opposite strand" is the right terminology for this point.

Data Availability: although it is understandable to not include the full WGS data it may be useful to take out the region that includes only the breakpoints for upload to a public database (such as SRA).

Reviewer #3 (Comments to the Authors (Required)):

Rodrigues and colleagues in "Single variant, yet "double trouble": TSC and KBG syndrome due to a large de novo inversion" report a patient with a large de novo pericentric inversion in chromosome 16 disrupting two genes, TSC2 and ANKRD11. The patient was initially diagnosed with tuberous sclerosis (TSC) and additional uncommon clinical traits with negative diagnostic results for TSC1 and TSC2.

This is a very interesting report about a large inversion causing two autosomal dominant diseases. My main concern regards the lack of detailed information about the tests performed in the patient, the superficial description of the breakpoint junctions, and very brief genotype-phenotype analysis. The authors propose a dual diagnosis for this patient, which is likely correct, but their data allow additional supportive evidence for such a hypothesis that they should take advantage of.

More specifically:

1- Methodology:

- previous tests performed on this patient were not well described. For instance, it is not clear why the TSC1 and TSC2 panel sequencing did not detect the inversion.
- WGS analysis is very superficial. They stated that "an unbiased singleton-WGS analysis was performed which did not identified any additional P/LP variants", but it is not clear how which program/platform they have used to annotate SNVs and Indels and rule out other genes contributing to the phenotype. Moreover, no SV caller was described.

2- Results:

- Did the WGS data provide split-reads? Inversions have two breakpoint junctions in cis but only one jct was shown in Figure 3 from the Sanger data. Please add a figure with the breakpoint junctions of both sides of the inversion. This information can be extracted from split-reads using IGV. Breakpoint junction analysis will enable confirmation that this is not a complex rearrangement accompanied by CNVs or that there are no insertions in either side. Then please compare WGS junctions to the Sanger sequencing junctions (P4/P2 and P3/P1) to confirm they are the same.

- Are junctions mapping to repeats (Alu, LINES, HERVs, etc)?

The fact that 5-25% of the TSC patients remain without a genetic diagnosis is interesting and raises the question whether copy-

neutral inversions contribute to this "missing heritability". If repeats are mediating the inversion formation by non-allelic homologous recombination, recurrence is possible.

- Was the inversion confirmed by FISH or karyotyping?

- Please add a figure indicating which transcripts of both genes were analyzed and are likely to be disrupted with exonic information and junction mapping. Each one of the genes has a few transcripts, so please specify. It will be helpful to have the inversions from other literature studies showing breakpoint location in the same transcript for comparison. How does the phenotype of the patients with inversions affecting TSC2 compares to the phenotype of the patient reported here?

- Please establish a genotype-phenotype for ANKRD11. How does the phenotype of this patient compare to other KBG patients? How about other patients with inversions? ANKRD11 is mainly associated with KBG syndrome but has also been identified in individuals with Cornelia de Lange syndrome (CdLS) and other developmental disorders caused by variants affecting different chromatin regulators.

- Would be possible to check RNA expression for each of the genes to confirm lack of transcription?

Other points:

1- Are the authors submitting their findings de-identified to ClinVar or breakpoint junctions to SRA (<https://www.ncbi.nlm.nih.gov/sra>)?

Reviewer #1 (Comments to the Authors (Required)):

Authors Rodrigues Alves Barbosa Victoria et. al. reports a patient with a clinical diagnosis of Tuberous Sclerosis (TSC) and no molecular diagnosis from standard of care genetic testing. They performed short-read whole genome sequencing to identify an 87Mb pericentric inversion with breakpoints disrupting two known autosomal dominant disease genes TSC2 and ANKRD11.

This work is important in highlighting the challenges and limitations in the current standard of care genomic testing and the potential for finding novel variants expanding the mutational spectrum of previously known disease genes.

A: We thank Reviewer #1 for these comments.

Comments to authors:

Authors mention that they first performed a targeted analysis in the known genes for TSC and later an unbiased analysis of all SNVs and SVs (page 7) that resulted in a single variant relevant to patient's phenotype which is the same inversion. It will be useful to describe and discuss the prioritization workflow and the number of variants they analyzed to get to a single variant explaining the phenotype.

A: We have added more details in the Materials and Methods section (Lines 349-377) as well as in the Results section (Lines 165-171).

Results: "To be certain that the inversion is the best possible explanation for the phenotype in this family, we also performed unbiased singleton-WGS analysis where we considered entire genome and SNVs and SVs. Overall, after automatic filtration of polymorphisms (population frequency > 1%), low quality variants and low effect variants, we manually filtered out 1,433 coding or splicing SNVs/indels (including 1,420 heterozygous and 13 homozygous) and 135 SVs based on population frequencies, predicted effect, pathogenicity and gene-phenotype or gene-disease associations."

Methods: "The first stage of data analyses for patients with well-defined clinical features is gene-centric (Maroilley et al, 2022b), guided by a hypothesis and focused on a single locus (Shu et al, 2023). After quality check with FASTQC (Andrews S., 2010) and trimming with Trimmomatic v0.35 (Bolger et al, 2014), reads were aligned along the human reference genome GRCh37 with BWA-MEM v0.7.15 (Li, 2013), sorted by coordinates using Samtools v1.3.1 (Li et al, 2009) and marked for duplicates using MarkDuplicates Picard tool (Picard Tools - By Broad Institute).

We first called SNVs and indels using Freebayes v1.3.1 (Garrison & Marth, 2012) and we looked for rare potentially pathogenic variants in both coding and non-coding regions of the loci of interest (TSC1 and TSC2). We then scrutinized the loci by performing a visual assessment using Integrative Genomics Viewer (IGV) (Robinson et al, 2011). We looked for signatures of short reads, combining split reads and changes in coverage, that would suggest the presence of breakpoints of structural variants or complex rearrangements disrupting either the TSC1 or TSC2 gene. We realigned reads around the breakpoints identified in TSC2 to uncover the structure of the variants and identify split reads that would guide the design of primers for PCR confirmation.

In addition, we performed unbiased genome-wide singleton-WGS analyses to exclude any potential variants not in TSC1 or TSC2 that could explain the phenotype; both de novo and inherited variants were considered, as well as coding/non-coding and SNVs and SVs using in-house designed pipelines (Maroilley et al, 2022b, 2022a).

In brief, we semi-automatically filtered SNVs and indels called by Freebayes according to population frequencies, predicted effect on genes and multiple pathogenicity scores as in Maroilley et al. (Maroilley et al, 2022b). We also called SVs, complex genomic rearrangements and, mobile element insertions using RUFUS (Ostrander et al, 2018) and GRIDSS v.2.10.2 (Cameron et al, 2017). We used in-house developed scripts to filter and combine calls from both tools as in Maroilley et al. (Maroilley et al, 2022a) We called repeat expansions with ExpansionHunter (Dolzhenko et al, 2019) and ExpansionHunter Denovo v.0.9.0 (Dolzhenko et al, 2020) as in Maroilley et al. (Maroilley et al, 2023). Finally, we explored large copy number variations using an in-house developed method based on the analysis of the coverage.”

Large inversions are challenging to detect given the rate of false positives from short-read sequencing data. This seems to be a SINE-mediated inversion (both breakpoints seem to fall in or near an Alu repeat) and the structural variants from short-read sequencing is known to be enriched for false positives involving repeat elements. Discussion around this will help the readers appreciate the challenges and opportunities in the improving the genome sequencing based diagnostic tests.

A: Indeed, this inversion is SINE-mediated. We have added this information in the Results section (Lines 205-208) and created a Supplementary Figure (Figure S2) to illustrate it. In addition, we discussed the opportunities of genome sequencing to detect such variant (Lines 282-289).

Results: “This breakpoint is 27 bp away from a SINE repetitive element (AluY; see Figure S2) and disrupts all isoforms of TSC2 (Figure S2) either of intron 15, 12 or 14. It overlaps a SINE repetitive element (AluSx; see Figure S2) and it disrupts all isoforms of ANKRD11 (Figure S2). This suggests that this large pericentric inversion is SINE-mediated.”

Discussion: “In this case, even though inversions are often due to repetitive regions and the known limitations of srWGS in properly genotyping those regions, a clear signature of inversion could be detected based on split reads at the breakpoint junctions and pair-mates aligned in irregular orientations. Such signatures can either be detected by manually exploring read alignments on a genomic viewer, but several tools are now able to report it. By improving genomic pipelines with tailored bioinformatics approaches, srWGS represent a great opportunity to detect such variants.”

Reviewer #2 (Comments to the Authors (Required)):

Summary:

Herein Victoria and Maroilley et al., present a case of a patient with an atypical presentation of Tuberous sclerosis (TSC). Short-read WGS uncovered a large 87 Mb pericentric inversion with breakpoints mapping to the genes TSC2 (explaining the TSC diagnosis) as well as ANKRD11 causing KBG syndrome suggesting a dual molecular diagnosis for this patient. Genomic inversions are a particularly cryptic form of structural variation as there is (typically) no copy number change associated with the event. Furthermore, breakpoints may lie within Repetitive or other difficult to away regions. This work is very well written and presents an important point that atypical presentations of disease may suggest a second molecular "hit" and more through sequencing methodologies (such as short-read WGS) may be required to uncover the full picture.

The conclusions regarding the molecular diagnosis of TSC and KBG is supported by breakpoint junction alignment showing disruption of both genes.

Major Comments

1. I would add more information relating to genomic inversion into the introduction to give the reader a better view of how this type of rearrangement is associated with human disease states. See: Feuk, Lars. 2010. "Inversion Variants in the Human Genome: Role in Disease and Genome Architecture." *Genome Medicine* 2 (2): 11.

A: More information about inversions added in the Introduction (Lines 91-114).

Introduction: "Chromosomal inversions were the first type of genetic variants to be studied, after being initially discovered in Drosophila (Sturtevant, 1917). Their impact could initially be considered as mild as there is no alterations in gene copy numbers, and in fact, human genome naturally carries multiple inversion. For instance, a 3.5 Mb inversion involving olfactory receptors on chromosome 8, was found present in 26% of healthy controls, having no clinical significance (Giglio et al, 2002). Pericentric inversions in chromosomes 1, 2, 3, 5, 9, 10 and 16 are amongst the most common in the human genome (Hsu et al, 1987). However, even though the DNA content may remain intact, the breakpoints of an inversion may cause serious gene disruptions, or impair transcription regulatory elements, leading to disease association (Feuk, 2010). Hemophilia A is one of the best-characterized examples, being caused by variants in chromosome X, which includes inversions present in 43% of patients (Lakich et al, 1993) It is also worth considering that inversions may also predispose the appearance of other minor variants in the offspring of their carriers (Sharp et al, 2006). Additionally, heterozygous inversions can impair chromosomal recombination and lead to differential gene expression patterns. They can affect both meiotic and somatic recombination and are also capable of inducing phenotypic variability in human diseases (Nomura et al, 2018). The effect of inversions is intricate and still not fully understood for a variety of reasons (Feuk, 2010). First, they are relatively rare, and multiple patients carrying the same inversion is often scarce. This is usually overcome when inversion is affecting a gene that has previous association with the disease. Second, rearrangements in DNA that do not cause any loss or gain of DNA sequence can hardly be detected with arrays, meaning that inversion mapping is mostly ineffective (Feuk, 2010). However, the advancement of sequencing technologies, such as paired-end

sequencing, allowed new strategies for the discovery of inversions (Tuzun et al, 2005)."

2. Posey et al., should be cited first as this was the major dual diagnosis paper as this (as far as I am aware) presents the concept more broadly to the field.

A: We have added Posey et al. first (Line 74).

3. "However, the diagnosis rate rarely reaches 50%, with still many patients going through an exhausting diagnostic odyssey" - Consider citing the following: Liu, Pengfei, Linyan Meng, Elizabeth A. Normand, Fan Xia, Xiaofei Song, Andrew Ghazi, Jill Rosenfeld, et al. 2019. "Reanalysis of Clinical Exome Sequencing Data." The New England Journal of Medicine 380 (25): 2478-80. Which shows that reanalysis can be a useful tool even for unresolved patients.

A: We have added this reference (Lines 65-66) in the Introduction.

Introduction: "Genetic dataset reanalysis, in this case, can also be of utmost help and has been proved to greatly increase molecular diagnosis."

4. diagnosis of more than one genetic disorder in an individual." I'm have not come across this term "double trouble" the literature and would stick to "dual molecular diagnosis".

A: "Double trouble" was previously used in the literature (see PMID 24070816). We have added that reference in the manuscript (Line 60).

Introduction: "It is also known as "double trouble" in the literature (Donkervoort et al, 2013)."

5. Complex rearrangements involving multiple inversions have also been associated with other Mendelian disease: Grochowski, Christopher M., Ana C. V. Krepischi, Jesper Eisfeldt, Haowei Du, Debora R. Bertola, Danyllo Oliveira, Silvia S. Costa, James R. Lupski, Anna Lindstrand, and Claudia M. B. Carvalho. 2021. "Chromoanagenesis Event Underlies a de Novo Pericentric and Multiple Paracentric Inversions in a Single Chromosome Causing Coffin-Siris Syndrome." Front. Genet. 12: 1478.

A: We have added more information regarding the diagnosis of CGRs and cited the above-mentioned paper (Lines 214-216).

6. What structural variant callers were run?

A: We have added more details in the Materials and Methods section (Lines 349-377).

Methods: "The first stage of data analyses for patients with well-defined clinical features is gene-centric (Maroilley et al, 2022b), guided by a hypothesis and focused on a single locus (Shu et al, 2023). After quality check with FASTQC (Andrews S., 2010) and trimming with Trimmomatic v0.35 (Bolger et al, 2014), reads were aligned along the human reference genome GRCh37 with BWA-MEM v0.7.15 (Li, 2013), sorted by coordinates using Samtools v1.3.1 (Li et al, 2009) and marked for duplicates using MarkDuplicates Picard tool (Picard Tools - By Broad Institute).

We first called SNVs and indels using Freebayes v1.3.1 (Garrison & Marth, 2012) and we looked for rare potentially pathogenic variants in both coding and non-coding regions of the loci of interest (TSC1 and TSC2). We then scrutinized the loci by performing a visual assessment using Integrative Genomics Viewer (IGV) (Robinson et al, 2011). We looked for signatures of short reads, combining split reads and changes in coverage, that would suggest the presence of breakpoints of structural variants or complex rearrangements disrupting either the TSC1 or TSC2 gene. We realigned reads around the breakpoints identified in TSC2 to uncover the structure of the variants and identify split reads that would guide the design of primers for PCR confirmation.

In addition, we performed unbiased genome-wide singleton-WGS analyses to exclude any potential variants not in TSC1 or TSC2 that could explain the phenotype; both de novo and inherited variants were considered, as well as coding/non-coding and SNVs and SVs using in-house designed pipelines (Maroille et al, 2022b, 2022a).

In brief, we semi-automatically filtered SNVs and indels called by Freebayes according to population frequencies, predicted effect on genes and multiple pathogenicity scores as in Maroille et al. (Maroille et al, 2022b). We also called SVs, complex genomic rearrangements and, mobile element insertions using RUFUS (Ostrander et al, 2018) and GRIDSS v.2.10.2 (Cameron et al, 2017). We used in-house developed scripts to filter and combine calls from both tools as in Maroille et al. (Maroille et al, 2022a) We called repeat expansions with ExpansionHunter (Dolzhenko et al, 2019) and ExpansionHunter Denovo v.0.9.0 (Dolzhenko et al, 2020) as in Maroille et al. (Maroille et al, 2023). Finally, we explored large copy number variations using a in-house developed method based on the analysis of the coverage.”

7. Are there other CNVs that may be present?

A: Our analysis revealed only two rare deletions (not present in population databases) and overlapping non-coding parts of genes for which the phenotype does not fit the patient's phenotype.

8. Was a chromosomal microarray or karyotype ever performed for this patient?

A: The patient had a microarray with no significant findings identified; the karyotype was not completed for this patient.

Figures:

9. Figure 1: it would be useful to include a depiction of the full chromosome 16 above the IGV view to show the exact regions that you are zooming into.

A: We have updated Fig. 1 accordingly.

10. Figure 2: feels unnecessary for a main text figure and would be better suited for a supplemental figure.

A: We have changed Figure 2 as Supplementary Figure S1.

11. Figure 3: this is a very informative and well laid out figure.

A: We thank the Reviewer for this comment.

12. It may be helpful to show the nucleotide level resolution of the breakpoints. Was it a blunt fusion? Microhomology?

A: Based on our analysis of the reads and the Sanger sequencing, it was more like a blunt fusion (Lines 208-212 and 228-233). We have added that information in the Results section and illustrated it in Figure 3.

Results: "One set of short split reads showed one of the breakpoints and the re-alignment of these reads along the reference genome (hg19) using Blat (Kent, 2002) showed no microhomology, but a hexamer CAGCCG that is present on both side of the junction in the reference genome, with a substitution T>G (reference being CAGCCT) at the 16q24.3 locus (Figure 2)."

"We analysed the breakpoints junctions using the Sanger sequences obtained (Figure 2). The breakpoint junction sequence on 16p13.3 shows a blunt fusion with a overlap of one nucleotide: a G that is present on both sides of the breakpoint in the reference genome. The breakpoint junction sequence on 16q24.3 showed a blunt fusion and no microhomology and confirmed the presence of the hexamer already observed in the srWGS read sequence (Figure 2)."

13. Are there genomic features in that region that may play a factor (Alus, LCRs, etc) in the formation of this inversion?

A: The inversion is SINE-mediated with Alu elements involved. We have added that information in the manuscript (Lines 205-208) and illustrated it in Supplementary Figure S2.

Results: "This breakpoint is 27 bp away from a SINE repetitive element (AluY; see Figure S2) and disrupts all isoforms of TSC2 (Figure S2) either of intron 15, 12 or 14. It overlaps a SINE repetitive element (AluSx; see Figure S2) and it disrupts all isoforms of ANKRD11 (Figure S2). This suggests that this large pericentric inversion is SINE-mediated."

14. Was any experimental validation performed to show disruption of these genes? A qPCR of cDNA for these genes should show an expression difference from normal controls if they are being disrupted by the inversion. A western blot would also nicely show a disruption of the genes.

A: No transcript level analyses were performed. Given the TSC clinical diagnosis and the fact that the identified inversion is the only variant of significance that we identified in this genome, we did not think that additional patient sampling is warranted.

Minor Comments:

15. I don't see "KBG" syndrome spelled out in the manuscript.

A: The explanation "KBG" refers to the surname initials of the three families originally diagnosed with the syndrome." was added on lines 172-173.

"KBG" refers to the surname initials of the three families originally diagnosed with the syndrome (Herrmann, 1975)."

16. Please include Chr when mentioning genomic locations so "Chr16:2,115,464"

A: "Chr" was added before chromosomal coordinates on lines 168, 170 and 708.

17. "The second breakpoint is on the opposite strand..." I don't think "the opposite strand" is the right terminology for this point.

A: We have removed that statement (Line 170).

18. Data Availability: although it is understandable to not include the full WGS data it may be useful to take out the region that includes only the breakpoints for upload to a public database (such as SRA).

A: We have now added the sequence of the region around the breakpoints from the srWGS reads and the Sanger sequencing in Figure 3.

Reviewer #3 (Comments to the Authors (Required)):

Rodrigues and colleagues in "Single variant, yet "double trouble": TSC and KBG syndrome due to a large de novo inversion" report a patient with a large de novo pericentric inversion in chromosome 16 disrupting two genes, TSC2 and ANKRD11. The patient was initially diagnosed with tuberous sclerosis (TSC) and additional uncommon clinical traits with negative diagnostic results for TSC1 and TSC2.

This is a very interesting report about a large inversion causing two autosomal dominant diseases. My main concern regards the lack of detailed information about the tests performed in the patient, the superficial description of the breakpoint junctions, and very brief genotype-phenotype analysis. The authors propose a dual diagnosis for this patient, which is likely correct, but their data allow additional supportive evidence for such a hypothesis that they should take advantage of.

More specifically:

1- Methodology:

- previous tests performed on this patient were not well described. For instance, it is not clear why the TSC1 and TSC2 panel sequencing did not detect the inversion.

A: The clinical *TSC1* and *TSC2* panel sequencing only focused on SNVs, so the inversion was missed.

2- WGS analysis is very superficial. They stated that "an unbiased singleton-WGS analysis was performed which did not identified any additional P/LP variants", but it is not clear how which program/platform they have used to annotate SNVs and Indels and rule out other genes contributing to the phenotype. Moreover, no SV caller was described.

A: We have added more details in the Materials and Methods section (Lines 349-377) as well as in the Results section (Lines 165-171).

Results: "To be certain that the inversion is the best possible explanation for the phenotype in this family, we also performed unbiased singleton-WGS analysis where we considered entire genome and SNVs and SVs. Overall, after automatic filtration of polymorphisms (population frequency > 1%), low quality variants and low effect variants, we manually filtered out 1,433 coding or splicing SNVs/indels (including 1,420 heterozygous and 13 homozygous) and 135 SVs based on population frequencies, predicted effect, pathogenicity and gene-phenotype or gene-disease associations. We also considered compound heterozygous as a potential genetic origin."

Methods: "The first stage of data analyses for patients with well-defined clinical features is gene-centric (Maroille et al, 2022b), guided by a hypothesis and focused on a single locus (Shu et al, 2023). After quality check with FASTQC (Andrews S., 2010) and trimming with Trimmomatic v0.35 (Bolger et al, 2014), reads were aligned along the human reference genome GRCh37 with BWA-MEM v0.7.15 (Li, 2013), sorted by coordinates using Samtools v1.3.1 (Li et al, 2009) and marked for duplicates using MarkDuplicates Picard tool (Picard Tools - By Broad Institute). We first called SNVs and indels using Freebayes v1.3.1 (Garrison & Marth, 2012) and we looked for rare potentially pathogenic variants in both coding and non-coding regions of the loci of interest (TSC1 and TSC2). We then scrutinized the loci by performing a visual assessment using Integrative Genomics Viewer (IGV) (Robinson et al,

2011). We looked for signatures of short reads, combining split reads and changes in coverage, that would suggest the presence of breakpoints of structural variants or complex rearrangements disrupting either the TSC1 or TSC2 gene. We realigned reads around the breakpoints identified in TSC2 to uncover the structure of the variants and identify split reads that would guide the design of primers for PCR confirmation.

In addition, we performed unbiased genome-wide singleton-WGS analyses to exclude any potential variants not in TSC1 or TSC2 that could explain the phenotype; both *de novo* and inherited variants were considered, as well as coding/non-coding and SNVs and SVs using in-house designed pipelines (Maroille et al, 2022b, 2022a).

In brief, we semi-automatically filtered SNVs and indels called by Freebayes according to population frequencies, predicted effect on genes and multiple pathogenicity scores as in Maroille et al. (Maroille et al, 2022b). We also called SVs, complex genomic rearrangements and, mobile element insertions using RUFUS (Ostrander et al, 2018) and GRIDSS v.2.10.2 (Cameron et al, 2017). We used in-house developed scripts to filter and combine calls from both tools as in Maroille et al. (Maroille et al, 2022a) We called repeat expansions with ExpansionHunter (Dolzhenko et al, 2019) and ExpansionHunter Denovo v.0.9.0 (Dolzhenko et al, 2020) as in Maroille et al. (Maroille et al, 2023). Finally, we explored large copy number variations using a in-house developed method based on the analysis of the coverage.”

Results:

3- Did the WGS data provide split-reads? Inversions have two breakpoint junctions in cis but only one jct was shown in Figure 3 from the Sanger data. Please add a figure with the breakpoint junctions of both sides of the inversion. This information can be extracted from split-reads using IGV. Breakpoint junction analysis will enable confirmation that this is not a complex rearrangement accompanied by CNVs or that there are no insertions in either side. Then please compare WGS junctions to the Sanger sequencing junctions (P4/P2 and P3/P1) to confirm they are the same.

A: Yes the WGS provided split reads. We used the split reads as a first line of *in silico* confirmation by locally re-aligning them (using UCSC Blat). We then use the split-read sequences to guide the design of the primers for further experimental validation.

We have added this information in the manuscript (Lines 208-212 and 228-233) and in Figure 3.

Results: “One set of short split reads showed one of the breakpoints and the re-alignment of these reads along the reference genome (hg19) using Blat (Kent, 2002) showed no microhomology, but a hexamer CAGCCG that is present on both side of the junction in the reference genome, with a substitution T>G (reference being CAGCCT) at the 16q24.3 locus (Figure 2).”

“We analysed the breakpoints junctions using the Sanger sequences obtained (Figure 2). The breakpoint junction sequence on 16p13.3 shows a blunt fusion with an overlap of one nucleotide: a G that is present on both sides of the breakpoint in the reference genome. The breakpoint junction sequence on 16q24.3 showed a blunt fusion and no microhomology and confirmed the presence of the hexamer already observed in the srWGS read sequence (Figure 2).”

4- Are junctions mapping to repeats (Alu, LINES, HERVs, etc)?

A: The inversion is SINE-mediated with Alu elements involved. We have added that information in the manuscript (Lines 205-208) and illustrated it in Supplementary Figure S2.

Results: "This breakpoint is 27 bp away from a SINE repetitive element (AluY; see Figure S2) and disrupts all isoforms of TSC2 (Figure S2) either of intron 15, 12 or 14. It overlaps a SINE repetitive element (AluSx; see Figure S2) and it disrupts all isoforms of ANKRD11 (Figure S2). This suggests that this large pericentric inversion is SINE-mediated."

5- The fact that 5-25% of the TSC patients remain without a genetic diagnosis is interesting and raises the question whether copy-neutral inversions contribute to this "missing heritability". If repeats are mediating the inversion formation by non-allelic homologous recombination, recurrence is possible.

A: We add a comment in the Discussion (Lines 239-240).

"Our work shows that undetected balanced events such as inversion could contribute to the missing heritability in TSC cases."

6- Was the inversion confirmed by FISH or karyotyping?

A: The patient had a microarray with no significant findings identified; however, neither karyotyping nor FISH was completed for this patient.

7- Please add a figure indicating which transcripts of both genes were analyzed and are likely to be disrupted with exonic information and junction mapping. Each one of the genes has a few transcripts, so please specify.

A: We have added this information in the manuscript and in Figure S2.

8 - It will be helpful to have the inversions from other literature studies showing breakpoint location in the same transcript for comparison. How does the phenotype of the patients with inversions affecting TSC2 compares to the phenotype of the patient reported here?

A: Unfortunately, there is no (or very limited) clinical information about the two other patients that have been reported with TSC2 inversions in the literature. Our patient has a clear clinical diagnosis of TSC with multiple manifestations of TSC (CNS, ophthalmologic, cardiac, cutaneous, neurodevelopmental).

9 - Please establish a genotype-phenotype for ANKRD11. How does the phenotype of this patient compare to other KBG patients? How about other patients with inversions? ANKRD11 is mainly associated with KBG syndrome but has also been identified in individuals with Cornelia de Lange syndrome (CdLS) and other developmental disorders caused by variants affecting different chromatin regulators.

A: We were unable to identify other patients described with *ANKRD11* inversion. This patient's phenotype is not in keeping with CdLS. However, the patient does have some features in keeping with KBG syndrome which include relative short stature, dysmorphic features (brachycephaly, tall forehead, wide central incisors, triangular face, broad eyebrows), and history of hypotonia. These findings are not

explained by TSC diagnosis. KBG is highly variable so while the patient does not have all classic findings of KBG syndrome, the phenotype is well within what has been reported in other affected patients.

10 - Would be possible to check RNA expression for each of the genes to confirm lack of transcription?

A: Given the TSC clinical diagnosis and the fact that the identified inversion is the only variant of significance that we identified in this genome, we did not think that additional patient sampling was warranted.

Other points:

11- Are the authors submitting their findings de-identified to ClinVar or breakpoint junctions to SRA (<https://www.ncbi.nlm.nih.gov/sra>)?

A: The junctions have been added to the manuscript (see Figure 2). The inversion will be added to ClinVar/dbVar.

December 18, 2023

RE: Life Science Alliance Manuscript #LSA-2023-02115-TR

Dr. Maja Tarailo-Graovac
University of Calgary
Biochemistry & Molecular Biology
3330 Hospital Drive NW
Calgary, Alberta T2N 4N1
Canada

Dear Dr. Tarailo-Graovac,

Thank you for submitting your revised manuscript entitled "Single variant, yet "double trouble": TSC and KBG syndrome due to a large de novo inversion". We would be happy to publish your paper in Life Science Alliance pending final revisions necessary to meet our formatting guidelines.

- please add ORCID ID for the secondary corresponding author--they should have received instructions on how to do so
- please add the Twitter handle of your host institute/organization as well as your own or/and one of the authors in our system
- the full name (first, middle name as initials, and the last name) of each author should be given on the title page
- please incorporate any points from the Conclusion section into the Results and Discussion
- please include ClinVar accession information in the Data Availability statement

A. FINAL FILES:

B. MANUSCRIPT ORGANIZATION AND FORMATTING:

**Submission of a paper that does not conform to Life Science Alliance guidelines will delay the acceptance of your

manuscript.**

The license to publish form must be signed before your manuscript can be sent to production. A link to the electronic license to publish form will be available to the corresponding author only. Please take a moment to check your funder requirements.

Sincerely,

January 16, 2024

RE: Life Science Alliance Manuscript #LSA-2023-02115-TRR

Dr. Maja Tarailo-Graovac
University of Calgary
Biochemistry & Molecular Biology
3330 Hospital Drive NW
Calgary, Alberta T2N 4N1
Canada

Dear Dr. Tarailo-Graovac,

Thank you for submitting your Research Article entitled "Single variant, yet "double trouble": TSC and KBG syndrome due to a large de novo inversion". It is a pleasure to let you know that your manuscript is now accepted for publication in Life Science Alliance. Congratulations on this interesting work.

DISTRIBUTION OF MATERIALS:

Again, congratulations on a very nice paper. I hope you found the review process to be constructive and are pleased with how the manuscript was handled editorially. We look forward to future exciting submissions from your lab.

Sincerely,
